# Role of Machine Learning in Precision Oncology: Applications in Gastrointestinal Cancers

**DOI:** 10.3390/cancers15010063

**Published:** 2022-12-22

**Authors:** Azadeh Tabari, Shin Mei Chan, Omar Mustafa Fathy Omar, Shams I. Iqbal, Michael S. Gee, Dania Daye

**Affiliations:** 1Department of Radiology, Massachusetts General Hospital, 55 Fruit Street, Boston, MA 02114, USA; 2Harvard Medical School, Boston, MA 02115, USA; 3Yale University School of Medicine, 330 Cedar Street, New Haven, CT 06510, USA; 4Center for Vascular Biology, University of Connecticut Health Center, Farmington, CT 06030, USA

**Keywords:** radiomics, machine learning, abdominal cancer, integrated multi-omics, precision oncology

## Abstract

**Simple Summary:**

Worldwide gastrointestinal (GI) malignancies account for about 25% of the global cancer incidence. For some malignancies, screening programs, such as routine colon cancer screenings, have largely aided in the early diagnosis of those at risk. However, even after diagnosis, many GI malignancies lack robust biomarkers to serve as definitive staging and prognostic tools to aid in clinical decision-making. Radiomics uses high-throughput data to extract various features from medical images with the potential to aid personalized precision medicine. Machine learning is a technique for analyzing and predicting by learning from sample data, finding patterns in it, and applying it to new data. We reviewed the fundamental concepts of radiomics such as imaging data acquisition, lesion segmentation, feature design, and interpretation specific to GI cancer studies and assessed the clinical applications of radiomics and machine learning in diagnosis, staging, evaluation of tumor prognosis, and treatment response.

**Abstract:**

Gastrointestinal (GI) cancers, consisting of a wide spectrum of pathologies, have become a prominent health issue globally. Despite medical imaging playing a crucial role in the clinical workflow of cancers, standard evaluation of different imaging modalities may provide limited information. Accurate tumor detection, characterization, and monitoring remain a challenge. Progress in quantitative imaging analysis techniques resulted in ”radiomics”, a promising methodical tool that helps to personalize diagnosis and treatment optimization. Radiomics, a sub-field of computer vision analysis, is a bourgeoning area of interest, especially in this era of precision medicine. In the field of oncology, radiomics has been described as a tool to aid in the diagnosis, classification, and categorization of malignancies and to predict outcomes using various endpoints. In addition, machine learning is a technique for analyzing and predicting by learning from sample data, finding patterns in it, and applying it to new data. Machine learning has been increasingly applied in this field, where it is being studied in image diagnosis. This review assesses the current landscape of radiomics and methodological processes in GI cancers (including gastric, colorectal, liver, pancreatic, neuroendocrine, GI stromal, and rectal cancers). We explain in a stepwise fashion the process from data acquisition and curation to segmentation and feature extraction. Furthermore, the applications of radiomics for diagnosis, staging, assessment of tumor prognosis and treatment response according to different GI cancer types are explored. Finally, we discussed the existing challenges and limitations of radiomics in abdominal cancers and investigate future opportunities.

## 1. Introduction

Worldwide gastrointestinal (GI) malignancies affect up to 4.8 million people per year, accounting for about 25% of the global cancer incidence [1]. Despite advancements made in understanding GI cancers, colorectal cancer remains the second most common cause of cancer deaths globally [2]. For some malignancies, screening programs, such as routine colon cancer screenings, have largely aided in the early diagnosis of those at risk [3]. However, some GI malignancies do not have effective screening tests and are extremely difficult to discern in the early phase, such as pancreatic cancer [4]. Furthermore, diagnosis of GI malignancies is often invasive, requiring biopsy and pathologic analysis following surgical resection. Even after diagnosis, many GI malignancies lack robust biomarkers to serve as definitive staging and prognostic tools to aid in clinical decision-making [5].

As medical imaging advanced, so too has image interpretation, particularly computer-assisted analysis. First pioneered by Philippe Lambin in 2012, radiomics uses high-throughput data to extract various features from medical images with the potential to aid personalized precision medicine [6]. With the evolution of artificial intelligence (AI), this field has grown rapidly and is being widely used in oncology [7]. Machine learning (ML), which is strictly associated with AI, is a general concept indicating the ability of a machine in learning and, thus, improving patterns and models of analysis [8,9]. In oncology, by extracting certain features from medical images and translating them into quantitative data for analysis, radiomics provides a noninvasive and efficient method for diagnosis, classification, and differentiation of lesions, tumor subtypes, and prognosis prediction in patients undergoing treatment [10,11].

Here, we reviewed the fundamental concepts of radiomics, such as imaging data acquisition, lesion segmentation, feature design, and interpretation specific to GI cancer studies. We also assessed the clinical applications of radiomics and ML in diagnosis, staging the evaluation of tumor prognosis and treatment response. Finally, we discussed the current challenges and limitations of radiomic, and investigate their future applications in GI cancers.

## 2. Methodology of Radiomics Extraction in Abdominal Cancer

### 2.1. Data Acquisition and Curation

Radiomics pipelines start with the acquisition of medical images. The most commonly used imaging modality is CT, followed by MRI and positron emission tomography (PET) [12]. When acquiring data using CT, critical parameters such as variations of Hounsfield Units (HU), density, contrast resolution, and pitch are all critical factors. The signal intensity of CT imaging allows for a direct correlation with tissue density. Slice thickness is also an important parameter; the thickness of each image affects photon statistics and, potentially, kilovoltage peak [13]. For most GI stromal tumors, CT imaging has been used for radiomic feature extraction, with images being acquired in the venous (50%) and arterial phases (40%) for analysis. Limitations of CT imaging for radiomic feature extraction mainly include reproducibility [14]. PET is another imaging modality that is commonly used in the workup of GI cancers. Similar to CT, the voxels in PET scans have quantitative properties. Challenges with PET arise in the standardization of PET protocols across and even within institutions. Inherent issues with PET protocols arise given the nature of imaging acquisition, as a multitude of factors can include the standard uptake value. These may be physiologic, including patient motion, inflammation, or blood glucose levels. They may also be technical, including differences in calibration threshold, synchronization, injection time, and method of delivery [15]. Specific to GI cancers, data acquisition may be limited by radiopharmaceuticals used for certain GI cancers (e.g., DOTATOC for neuroendocrine tumors), in addition to the percentage threshold of the maximum standard uptake value used to delineate the tumor of interest [16]. Figure 1 demonstrates the flowchart of the application of AI in radiology for GI cancers [17,18].

Compared with CT and PET, the voxel values from MRI have limited quantitative value, as they are influenced by a variety of intrinsic and extrinsic elements. As for MRI, in a seminal paper from Buch et al., the group demonstrates great variety in texture features when different MRI acquisitions were analyzed [19]. One group of GI cancers primarily assessed with MRI is rectal cancer; the gold standard for local staging is completed using MRI, particularly high-spatial-resolution T2-weighted (T2W) imaging, as it demonstrates critical anatomic landmarks of relevant structures. Acquisition parameters specifically relevant for MRI-trained radiomic algorithms include the scanner type, field-of-view, repetition time, inversion time, echo train length, number of signals averaged, and the spacing of pixels [20,21].

The information mined from the images contributes to a large pool of data that can include disease outcomes, treatment time points, pathology information, genomic data, or other clinical features which are used to create descriptive and predictive models in clinical patient care. Taking into consideration the impact of inconsistent imaging acquisition protocols and reconstruction methods across imaging centers and different manufacturers, imaging data need to be preprocessed [22]. Commonly used procedures include resampling and intensity normalization. Image resampling is applied for image quality improvement and eliminates imaging resolution non-uniformity. The normalization of image intensity improves the intensity variations between the subjects by transforming all images from their original greyscale into a standard greyscale. In a previous study, liver signal intensity was normalized depending on the spleen signal intensity on hepatobiliary phase images and demonstrated better diagnostic value when compared with non-normalized data [23].

### 2.2. Segmentation

Segmentation of the region of interest (ROI) (2D) or of the volume of interest (VOI) (3D) is the next step in the radiomics analysis process. This step delineates the image components to be included in the analysis and used in the model. Segmentation can be divided into manual, semiautomatic, and automatic processes. Radiologists performed manual segmentation to annotate the location and precise lesion margin in most radiomics studies [24,25]. Another method of manual segmentation occurs by placing a rectangular/circle box via deep learning (DL) analysis. Considering the possible intra-reader variability and subjective judgement in manual segmentation, segmentations by multiple clinicians, at different time points, are required to decrease the intra- and inter-reader variability.

Automated image segmentation, though still in its infancy, is currently being explored as a promising method to segment thousands of images with low error; however, many algorithms are not yet fully generalizable [26]. In automatic segmentation, ROI annotation is done using ML, whereas semiautomatic segmentation still requires partial manual intervention before automatic segmentation can take place [9]. A study by Men et al. showed that fully automated systems, such as a deep dilated convoluted neural network (CNN) based model, may provide even better reproducibility performance compared with U-Net methods in patients with rectal cancer [27]. The three categories of classic segmentation algorithms are based on: (a) intensity thresholds and regions (global/local thresholding, region growing and splitting, merging methods), (b) a statistical approach (parametric mapping and maximization segmentation algorithms) as well as clustering (k-means and fuzzy clustering) and deformable model approaches, and (c) an Artificial Neural Network and Atlas Guided Approach [28].

Segmentation algorithms may depend on the clinical endpoint of interest. For example, within gastric cancers, prediction of histological grade or tumor grading may require segmentation of entire tumors before surgery on arterial and portal imaging phases or on apparent diffusion coefficient maps, whereas segmentation for the purpose of predicting the outcomes of surgical resection may rely on volumetric segmentations [29,30]. Reproducibility and robustness are critical aspects of ROI segmentation in radiomics and are assessed through the calculation of intra-class correlation coefficients and concordance correlation coefficients. While inter-viewer and intra-viewer variability have been studied in the segmentation of other cancers, such as brain and lung cancer, it is of great current interest for GI cancers [12,31]. Wong et al. recently assessed interobserver and interdisciplinary (radiation oncology vs. radiology) agreement for tumor volumes in pancreatic cancer. They concluded that there were large variations of intraclass correlation coefficients within both groups, with radiation oncology having slightly higher stability in feature detection [32]. In gastric cancers, delta radiomic models using semi-automated segmentation DL algorithms have also been utilized to predict the response to chemotherapy for patients with advanced gastric cancer; notably, a semi-automatic segmentation method using a V-net CNN DL algorithm outperformed manual segmentation in reproducibility [33,34].

### 2.3. Feature Extraction

After the ROI is delineated, image characteristics are extracted. Manual engineered (shape/histogram/texture-based) and DL features are the two main types of radiomic features. Shape-based features describe the geometric attributes of the ROIs. Histogram features capture the first-order statistical characteristics of the organ or the lesion. Textural features, extracted from a series of high-order textural matrixes, outline the granular pattern of the ROIs [35,36]. Table 1 provides a broad overview of texture or radiomic features.

Shindo et al. used histogram analysis in a diffusion-weighted imaging (DW) MRI to differentiate pancreatic adenocarcinoma from neuroendocrine malignancies by assessing b-values; they demonstrated that the histologic entropy, skewness, and kurtosis of ADC values were higher in adenocarcinomas [37]. Rectal cancers, commonly diagnosed by MR, have also been assessed in a similar fashion; several recent studies have shown data extracted from DWI and T2W sequences may be useful in response to chemoradiotherapy [38,39]. In one notable radiomics study on a series of rectal cancer patients, data from MRI scans were extracted for tumor intensities, textual features based on heterogeneity within the segmented tumor, and textual features based on wavelet decompositions [40]. Indeed, automated segmentation, particularly using deep learning, has been emphasized as a key method for improving reproducibility and performance in addition to superiority in speed and time [41,42,43].

The DL network extracts supplementary high-dimensional features and encodes medical images into shape abstract textural information via shallow and deep layers, respectively. Wang et al. suggested a novel CNN-based method to extract DL features from MRI automatically. They reported that DL features performed superior to textural features in predicting malignancy in hepatic lesions [44].

## 3. Radiomics and Machine Learning in Diagnosis and Staging of GI Cancer

### 3.1. Gastric (Stomach) Cancer

Gastric cancer, often diagnosed at an advanced stage, has a poor prognosis and is often resistant to therapy [45]. Gastric cancer is characterized by substantial heterogeneity, which increases the chance of tumor relapse even after chemotherapy (CTx). Most of the radiomics studies on gastric cancer are focused on prognosis and therapeutic response. However, we found one study that focused on differential diagnosis. Gastric cancer can mimic other gastrointestinal tumors with remarkably different management and therapy—these include primary gastric lymphoma and stromal tumors [46]. It is difficult to differentiate these tumors based on imaging characteristics, and a biopsy is usually required. Radiomics analysis has shown promise in differentiating these tumors based on textural features. Ba-Ssalamach et al. used texture analysis from CT scans to differentiate gastric cancer from gastric lymphoma with a misclassification rate of only 3.1% [46].

### 3.2. Colorectal Cancer

The gold standard imaging modality for local staging and restaging after treatment is an MRI, which can detect high-risk prognostic factors in colorectal cancer (CRC). Computed tomography has been long-established to detect distant metastases [47]. However, CRC characterizations remain measurable only—after surgery and histopathology assessment. AI, radiomics, and ML are promising techniques that could further enhance the value of medical imaging in this cancer, allowing the design and implementation of decision-support tools based on quantitative data [7]. Hong et al. showed that a combined model based on pre-operative CT radiomics and CT staging significantly outperformed the CT staging-only model in detecting high-risk colon tumors [48]. In a study performed on 502 patients with CRC, the radiomics model based on portal-phase CT images achieved substantial diagnostic performance with 84% accuracy and an area under the curve (AUC; a two-dimensional area which has been calculated by using the integration formula) of 0.94 for differentiating hepatic lesions [49]. Moreover, radiomics provides a deep characterization of tumor phenotypes regarding the underlying pathophysiology or genetic changes by converting medical images into structural information and mineable data. CT-based radiomics has predicted the mutation status in patients with CRC and in lung adenocarcinoma patients for KRAS/BRAF and EGFR, respectively [50,51]. Several studies have shown that the combination of clinical and radiomics models achieved good performance in the prediction of MSI status in CRC patients [52,53,54]. A radiomics nomogram incorporating radiomics signatures and clinical indicators achieved an AUC of 0.77 when predicting the microsatellite instability (MSI) status [55]. In a recent study, Ying et al. reported that the combined model based on pre-operative CT radiomic features and clinical variables had an AUC of 0.90 in predicting the MSI status of patients with CRC [52].

Rectal cancer can be diagnosed with MRI, which can help identify patients who are suitable for chemoradiotherapy and surgery, in addition to looking at vascular invasion and spread to the mesorectal fascia [21]. Locally advanced rectal cancer (LARC) is most commonly studied; several studies have demonstrated value in utilizing T2-weighted sequences for the diagnosis of rectal cancer [56,57]. Additionally, assessment of radiomic features may also aid in the staging of rectal cancer; using 119 rectal cancer patients, Sun et al. created a model of MRI-derived characteristics that identified T stage with an AUC of 0.852 [58]. Lymph node analysis, allowing for N staging, has also been explored using MRI imaging. Some groups have been able to achieve algorithms to discriminate N0 from N1–2 patients with moderately strong sensitivities and specificities, in addition to predicting nodal pathology following neoadjuvant chemotherapy (nCRT) [57,59].

### 3.3. Pancreatic Cancer and Neuroendocrine Tumors

Pancreatic cancer is an insidious cancer that results in high rates of mortality, likely because early presentations are particularly difficult to detect on imaging [60]. Surgical resection is the only definitive treatment for pancreatic cancer. The most common subtype of pancreatic cancer, pancreatic adenocarcinoma, arises from pancreatic exocrine glands and accounts for more than 80% of pancreatic tumors. Most pancreatic cancers present in the head of the pancreas (60–75%) which affects symptomatology and surgical resectability [61]. In addition, it is still debated to what extent surgical resectability is predicted by other markers [62,63]. Notably, less than 20% of patients have resectable cancers at the time of diagnosis [64]. After surgical resection, it is usually necessary for patients to complete extensive neoadjuvant and adjuvant chemotherapy and radiotherapy (e.g., stereotactic body radiation therapy) to prevent a recurrence. Even in these cases, due to the changing tumor microenvironment, it is incredibly difficult to predict outcomes [65]. Similarly, the incidence of neuroendocrine tumors has paralleled advances made in imaging; over the past several decades, incidentally, the number of discovered pancreatic neuroendocrine tumors has increased, with the increased detection of tumors < 2 cm [66,67]. The first-line therapy for pancreatic neuroendocrine tumors is still surgery, although there are myriad therapies depending on various biomarkers and tumor grades. Preoperative appraisal of tumor grade has been demonstrated to be achievable with AUCs ranging from 0.7–0.9 using radiomic analyses [68].

Radiomics has been explored as a method to diagnose and stage pancreatic cancer. Given that occult pancreatic cancer is often not discernable on imaging until much later in the disease process, detection of lesions is not visible to the human eye, though advanced imaging and computational analysis techniques may facilitate earlier diagnosis and management. In a Taiwanese population, Chen and Chang et al. used an ML model using contrast-enhanced portal venous CT images to detect small (<2 cm) pancreatic ductal adenocarcinomas. In this cohort, the authors demonstrated sensitivities of 94.7% and 80.6% when used on Taiwanese and U.S. patient data sets [69]. Staging pancreatic cancer is also clinically challenging, but it is important in determining surgical candidates and adjuvant and neoadjuvant treatment regimens [70]. Various groups have created models to stage pancreatic cancer with varying degrees of accuracy [71,72]. A study conducted by An et al. utilized ML to predict lymph node metastasis for pancreatic adenocarcinoma. The Resnet 18 convolutional neural network was used to classify tumors into lymph nodes, positive or negative. A clinical model was created as well as the DL model. The AUC for the DL models outperformed the clinical model [73].

### 3.4. Liver Cancer

Radiomics enables non-invasive differentiation of focal liver lesions, the most common primary hepatic malignancy being HCC but also including hemangioma and metastases. CT is the most useful for imaging and grading liver cancer, specifically HCC. The pre-contrast and portal phase CT have been shown to be effective at differentiating HCC and non-HCC [74,75]. Radiomics signatures based on T2W-derived texture features of focal hepatic lesions can help classify hepatic hemangioma, hepatic metastases, and HCC with good diagnostic performances (AUC: 0.83–0.91) [76]. A previous study reported that primary liver tumors could be differentiated from metastatic lesions with an accuracy of 83% using 3D CNN features extracted from DWI images [77]. Lastly, ultrasound image analysis can also classify benign and malignant focal liver lesions (AUC: 0.94) and malignant subtyping (AUC: 0.97) [78].

### 3.5. GI Stromal Tumors

GI stromal tumors (GIST) are another subset of GI tumors that have been at the forefront of radiomics interest. Traditionally, predicting the behavior of GI stromal tumors on imaging is difficult, as they are often (1) indiscernible in the early stages and (2) when seen in imaging, they have already metastasized to distant locations [61]. Some models have assessed the ability to differentiate GISTs from non-GISTs to a good extent (AUC = 0.77), in addition to differentiating GISTs from other gastric cancers, such as adenocarcinomas and lymphomas [46,61,79]. Staging of GISTs is also of great interest; risk stratification using radiomic data may aid in discerning those amenable to surgery and preoperative risk [39].

## 4. Radiomics and Machine Learning in Prognosis and Treatment Response Prediction

### 4.1. Gastric Cancer

Gastric cancer recurs after nCRT in up to 30–40% of patients within 5 years [80,81]. Given this frequent recurrence, there is growing interest in predicting and monitoring treatment efficacy. A recent study by Cui et al. developed a radiomics nomogram that demonstrated satisfactory performance in predicting prognosis and response to nCRT, with an AUC of 0.829 and 0.827 in the training and validation cohorts [45]. Wang et al. applied a radiomics model to extract diagnostically relevant RFs to predict patients’ response to nCRT at the time of diagnosis [82]. Another study by Shin et al. successfully predicted the prognosis of recurrence-free survival using only pre-operative CT scans [83].

Vascular invasion holds an unfavorable prognosis in gastric cancer. Unfortunately, there are no reliable methods for the preoperative assessment of vascular invasion. Yang et al. developed and validated PET-CT-based radiomics signatures for predicting vascular invasion preoperatively. This study, and others assessing the efficacy of radiomics approaches in other types of cancer, suggests that PET/CT-based radiomics analysis might serve as a valuable tool for predicting vascular invasion and lymph node involvement in patients with gastric cancer [84,85].

Lastly, due to late diagnosis, gastric cancer is often metastasized, particularly in the peritoneal cavity, at the time of detection [86]. Thus, assessing peritoneal involvement early and accurately is critical for determining prognosis and optimal therapy. PET/CT is the main method for detecting peritoneal involvement. Xue et al. applied a radiomics model to predict peritoneal involvement based on PET imaging (AUC = 0.86 and 0.87 in training and validation cohorts, respectively) [87]. Dong et al. developed a model to identify peritoneal metastasis in patients in a multicenter cohort, demonstrating an AUC of 0.947, 0.928, and 0.920 in the three validation cohorts [88].

### 4.2. Colorectal Cancer

#### 4.2.1. Evaluation of Tumor Vascular Invasion

A combined model, including MRI-based EMVI status and a radiomics score for the lymphovascular invasion (LVI)/perineural invasion (PNI) estimation in patients with CRC, showed significant predictive power. CT may also play a role and has been shown to predict LVI and PNI in rectal cancer [89]. Imaging features, such as pre- and early post-treatment MRI parameters assessing sphincter involvement and extramural vascular invasion (EMVI), have been shown to be associated with patient outcomes [90,91]. In rectal cancer, radiomic features extracted from a whole-tumor volume on T2W images have been shown to outperform the combination of T2 and DWI in evaluating complete response (CR) [92].

#### 4.2.2. Prediction of Treatment Efficacy and Prognosis

CT may be used to classify treatment response and prognosis to varying degrees. GR has been predicted using both contrast and non-contrast-enhanced CTs. A study that used CT-based radiomics for the prediction of CR demonstrated that, while incorporating the same initial features, an SVM model outperformed the deep neural network [34]. Multi-modal models have also been used: PET/MRI and CT/MRI. The PET/MRI model performance was similar to the PET model but yielded better performance than the MRI-only model [93]. The CT/MRI outperformed the CT-only (AUC 0.91 vs. AUC 0.78, respectively) but was comparable with the performance of individual MRI sequences. In addition to treatment response, prognosis has been assessed. On MRI, multiple histograms, GLCM, and gray level run length matrix (GLRLM) features were correlated with disease-free survival [94,95]. CT has also been utilized; in a study by Dai et al., radiomics signatures were developed to predict recurrence-free and overall survival [96]. Some studies reported heterogeneous primary tumors (i.e., higher entropy and lower uniformity) are correlated with better OS, while other studies showed more homogeneous tumors are associated with improved disease-free or progression-free survival [5,97].

MRI is another modality to predict treatment efficacy and prognosis. Contrast-enhanced MRI may be more predictive than non-contrast MRI. The most frequently used modalities in radiomic studies that focus on response prediction of the primary tumor were T2W and DWI MRI [98]. Studies using ML classifiers, such as a support vector machine (SVM), random forest, and naive Bayesian network, resulted in promising results to predict pathological CR (AUC 0.71–0.87) [99,100]. MRI-based radiomics derived from T1W images of rectal cancer yielded moderate results to predict the pathological good response (GR), with an Ada boost classifier-based model outperforming a logistic regression model [101,102]. DWI-based imaging biomarkers have also been evaluated, including ADC, histogram features, and gray-level co-occurrence matrix (GLCM). These studies have demonstrated heterogeneity in predictability and usefulness for prognosticating CR and GR. Several studies have also combined multisequence models, which have outperformed classifiers for response prediction [5,103,104,105,106,107]. The field of radiomics applied to rectal cancer has mostly emphasized treatment after therapy, predominantly assessing locally advanced rectal cancer using T2w MRI and diffusion-weight MRI [108]. The study by Giraud et al. examined 2-year disease recurrence of anal squamous cell carcinoma using logistic regression. The mixed radiomic and clinical model outperformed the clinical model in the testing cohort, with an AUC of 0.898 compared with an AUC of 0.714 [109].

#### 4.2.3. CRC Metastases

CT-based radiomics may predict a response to CTx in colorectal liver metastasis (CRLM). Several studies have revealed that the predictive value of radiomics features is dependent on treatment, including whether patients received monoclonal antibodies [35,94]. Using MRI, Shi et al. reported higher histogram variance and lower GLCM uniformity on T2W images in responsive tumors [110]. Survival has also been assessed to varying degrees. Some studies have reported an association between OS and the AUC of the cumulative standard uptake value-volume histogram using 18F-FDG-PET/CT [111,112,113]. Simpson et al. found a lower texture signal was correlated with better OS of patients after hepatic surgery [114].

### 4.3. Pancreatic Cancer and Neuroendocrine Tumors

The prognosis of pancreatic cancer remains poor overall, with a five-year survival rate ranging from 5% to 15% [115]. The only definitive option is surgical resection, although only 20% of pancreatic cancers are amenable to resection by the time they are diagnosed [115]. Immunotherapies are also being increasingly explored, but this requires a detailed understanding of the tumor’s microenvironment and the ability to identify biomarkers such as PD-L1 expression, tumor-infiltrating lymphocytes, various genetic mutations, and immune checkpoints [116]. Although it is just in its infancy, radiomics may provide the ability to assist in the prognosis of pancreatic cancer. Various models have explored the ability to prognosticate in patients with pancreatic adenocarcinoma; this cancer proves more challenging to prognosticate using classic methods given its inherently poor prognosis [117,118]. Zhang et al. demonstrated a CNN-based approach can outperform conventional cox proportional hazard modeling in predicting survival patterns, although this method is still limited by relatively small sample sizes [118]. Using FDG-PET radiomics, another group created a model using a gray-level zone matrix and gray-level non-uniformity predictors to successfully stratify patients into three groups of poor prognoses [119]. Treatment response may also be determined by biomarkers, possibly predicted using radiomics or radiogenomics, which would allow for non-invasive and inexpensive surveillance. In one model, the authors demonstrate the AUC for radiogenomics-predicted p53 mutations to be 0.795, and that radiogenomic-predicted p53 mutations were associated with poor prognosis [120]. Numerous studies utilized radiomics to determine the prognosis and treatment response prediction in pancreatic adenocarcinoma [121,122,123,124,125]. The study by Nasief et al. utilized a Bayesian regularization backpropagation neural network to classify lesions into a good and poor response to treatment with an AUC of 0.92 [124]. The study by Mukherjee et al. looked at four ML models to classify lesions into normal or pre-diagnostic for pancreatic adenocarcinoma before a clinical diagnosis was made. These models included SVM, Random Forest, KNN, and XGBoost. SVM had an AUC of 0.98, Random Forests had an AUC of 0.95, KNN had an AUC of 0.95, and XGBoost had an AUC of 0.96. For reference, the radiologists who reviewed the images had an AUC of 0.66 [125].

For neuroendocrine tumors, predicting outcomes has proved more challenging given inherently smaller sample sizes [68,126]. Analysis of FDG PET/CT and Ga-DOTATOC have been used to predict angioinvasion, metastases, and tumor aggressiveness [127,128,129]. Few studies have demonstrated moderate performance using tumor heterogeneity to predict peptide receptor radionuclide therapy (PRRT) [130,131].

### 4.4. Liver Cancer

#### 4.4.1. Tumor Differentiation and Proliferation Measurements

After surgery, one of the risk factors of recurrent HCC with the highest importance is the histologic grade of the tumor [132]. Two recent studies investigated the potential of MRI-based radiomics as indicative biomarkers for HCC grade and aggressiveness characterization. They have shown the potential of radiomics [133,134]. In a recent prospective study, the tumor Ki-67 level could be assessed with good accuracy using pre-operative radiomics analysis [135].

#### 4.4.2. Evaluation of Tumor Vascular Invasion

It is critically important to detect microvascular invasion in HCC and differentiate neoplastic and bland portal vein thrombosis preoperatively [136,137]. It has been previously reported that the mean value of positive pixels and entropy can characterize portal vein thrombosis [138]. Recent studies have shown good diagnostic accuracy can be achieved using radiomic features extracted from CT for the prediction of microvascular invasion prior to surgery [139,140].

#### 4.4.3. Prediction of Treatment Efficacy and Prognosis

Previous studies have achieved an accurate prediction of prognosis and various therapy assessments by radiomics analysis [141,142]. Multiple studies performed liver resection evaluation, and one study was conducted for the assessment of liver transplantation [143,144,145,146,147,148]. Suh et al. reported that CT texture analysis can be helpful for prognosis prediction and effective treatment selection between transcatheter arterial chemoembolization and hepatic resection [149]. For HCC patients with prominent vascular invasion or extrahepatic spread, systematic treatment is the standard of care recommended by current guidelines [142,150]. A multicenter large study on advanced HCC revealed that entropy extracted from contrast-enhanced CT was associated with tumor heterogeneity, and entropy on portal venous phase images was an independent predictor for OS [151]. Emerging evidence from a retrospective multicohort study showed promising results in predicting immunotherapy response by combining CT-based radiomics and genomic data [152].

#### 4.4.4. Intrahepatic Cholangiocarcinoma (ICC)

Intrahepatic cholangiocarcinoma is an aggressive primary liver cancer originating from the bile duct epithelium; the only definitive cure is surgical resection [153]. Recent evidence revealed that early ICC recurrence after partial hepatectomy can be predicted with an AUC of 0.77 using radiomics on preoperative arterial-phase MR images [154]. Radiomics signature from portal venous phase CT has been shown to be predictive of lymph node metastasis in biliary tract cancers (AUC: 0.80) [155].

#### 4.4.5. Metastatic Hepatic Malignancies

A study by Lubner et al. showed that tumor grade, mutation status and overall survival were significantly associated with CT-derived texture features of CRLM prior to initiation of the treatment [156]. Another study by Beckers et al. found that the proportion between the lesion texture and the surrounding liver may reflect tumor aggressiveness, chemotherapy response, and OS [157]. Although, it has been reported that radiomics from liver parenchyma on portal venous phase CT cannot be used to predict the development of hepatic metastasis in patients with CRC [158]. In addition to colorectal cancer, CT texture features of esophagogastric liver metastases can help predict response to chemotherapy [159].

### 4.5. GI Stromal Tumors

Similarly, there is also interest in using radiomics to predict molecular or genetic features in order to aid with biologically targeted therapies for GISTs, although this too remains in infancy [79]. In terms of treatment for GISTs, various studies have employed texture analysis of CT to predict the preoperative outcomes of GISTs [160,161]. Radiogenomics is also being employed to assess Ki67, a tumor proliferative marker, as a prognostic indicator [162]. Lastly, rectal cancer will also benefit from radiomics-predicted treatment and prognostication. The current standard of care relies on imaging; although, given a heterogeneous patient population, personalized treatment schemes are difficult to achieve on standard imaging review.

Previous studies utilized radiomics in determining the prognosis and treatment response prediction in GI stromal tissues and anal squamous cell carcinoma. The study by Wang et al. examined the performance of various machine learning models for classifying gastrointestinal stromal tumors (GISTs) into high or low malignant potential [163]. Similar to the study conducted by Mukherjee et al., this paper used SVM, Random Forests and logistic regression. Random forests had the best performance with an AUC of 0.9, SVM yielded an AUC of 0.8, and logistic regression resulted in an AUC of 0.85. In the study conducted by Chen et al., a radiomics nomogram was created which also incorporated subjective CT findings and clinical indexes. These features were inputted into an SVM model which outperformed the traditional radiomics model with an AUC of 0.867 compared with an AUC of 0.858 [164].

## 5. Future Challenges and Opportunities

Despite considerable progress, there remain some challenges in the application of radiomics to GI malignancies. First, any AI or ML tool is only as strong as the training data provided. Predictive performance for automated tools remains limited by the lack of optimal thresholds necessary to balance sensitivity and specificity during data acquisition and curation [165,166]. Similarly, given that patient data is highly heterogenous due to age, sex, race, and demographics, future algorithm and machine learning technologies must account for such variations [167]. Furthermore, automated detection can still result in high rates of false positives. Conversely, radiomics holds the potential to aid in potential overdiagnosis when used in conjunction with other clinical tools [168,169,170]. It has been challenging to reach Dice similarity coefficients > mid 80% for segmentation accuracy using machine learning. Lastly, even the best machine learning systems do not yet perform at the level of a radiologist, and it is apparent there is room for more advancement. Testing the robustness of radiomics tools with prospective and retrospective real-life populations will be paramount to integrating such tools into clinical practice.

## 6. Conclusions

Radiomics as an emerging quantitative technique is growing rapidly in GI cancer management with consistently evolving methodology. The potential in diagnosis, treatment assessment and prognosis prediction in several GI cancers have been previously discussed despite the retrospective nature and single-modality basis of most of the studies. Therefore, further multicenter and prospective validation is still required to validate its clinical utility. The current main obstacles for the application of radiomics in cancer diseases consist of limited high-quality data collection and a lack of biological mechanistic explanation. Data sharing and collaborations on data cleaning and labelling across institutions may fill this gap. To facilitate broader translation and clinical adoption, more accurate and interpretable artificial intelligence algorithms need to be developed.

## Figures and Tables

**Figure 1 cancers-15-00063-f001:**
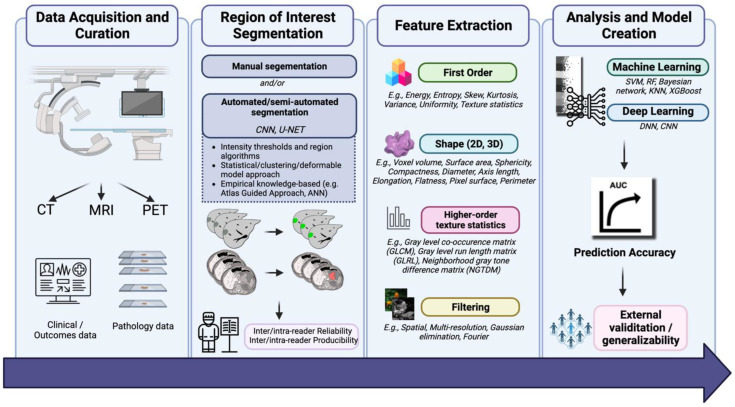
Flowchart of application of AI in radiology for GI cancers.

**Table 1 cancers-15-00063-t001:** Summary of quantitative features used in radiomics workflow.

First Order	Description
Energy	Magnitude of voxel values; also referred to as angular second moment or uniformity
Entropy	Randomness in image values
Skew	Quantifies asymmetry of distribution of a certain value
Kurtosis	Measures the “tailedness” of values relative to the mean
Variance	The squared deviation of a value
Uniformity	Sum of the squares of intensity values
**Shape Features**	
** *3D* **	
Volume Mesh Voxel	Can be calculated either using a mesh or without a mesh
Surface Area	Quantifies space surrounding the outside of region of interest
Sphericity	Assesses how similar the region of interest is to a sphere
Diameter	The Euclidean distance between two points in the region of interest, taking the shape mesh into account
Axis length	Distance between two points in the region of interest, regardless of the shape mesh
Elongation	Quantifies the length of the first two largest principal axes
Flatness	Quantifies the length of the largest and smallest principal axes
** *2D* **	
Area	Quantifies the space within a two dimensional region of interest
Perimeter	Quantifies the borders surrounding a two dimensional region of interest
Sphericity	Measures the similarity to a circle
Axis length	Distance between two points in the region of interest
Elongation	Quantifies the length of the first two largest principal axes
**Higher-order texture statistics**	
Gray level co-occurrence matrix (GLCM)	Quantifies pairs of pixels with certain gray level values
Gray level run length matrix (GLRL)	Quantifies the length of pixels within the same gray value, in 2 to 3 dimensions
Neighborhood gray tone difference matrix (NGTDM)	Quantifies the relationship between a pixel with surrounding gray level values
**Filtering**	
Spatial filtering	Based on neighborhood functions within the original image (examples: Gaussian, Laplacian, etc.)
Multi-resolution filtering	Based on variations in gray level differences within a region
Fourier transformations	Operation that converts a time/spatial signal to a frequency domain signal

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
