# Peer review of "Role of Machine Learning in Precision Oncology: Applications in Gastrointestinal Cancers"

_cancers, 2022, doi:10.3390/cancers15010063_

Round 1

Reviewer 1 Report

Dear Authors,

the topic of the article (Role of Radiomics Applications in Gastrointestinal Cancers) could be of great interest for the audience but there are many critical issues in the content and methods of your review.

First of all, GI cancers are a very heterogeneous group of tumors and expecially liver cancer and NETs can't be mixed with gastric, pancreatic, or stromal and anal cancer.

All the imaging techniques are put together in the article without any consideration for the specific characteristics and limits in radiomic analysis for each one.

For example, the problems of PET and CT or MR in extracting radiomic features are quite different. Expecially PET is approached only superficially without consider inherent problems and specific aspects due to the type of radiopharmaceuticals used for some cancers (i.e. DOTATOC for NETs) and the use of percentage threshold of the maximum SUV to delineate tumor.

Moreover, the technical aspects of radiomic features extraction are considered all together as the same thing i.e. algorythms, machine Learning and AI, but they are not.  

Finally, future applications are not investigated in the paper as you write in the abstract and  in the introduction.

So that, if you would revise the article it's necessary to radically change strategy. Some examples are given below: 

- to analyze only METHODS (segmentation, Data acquisition and curation, features extraction) to obtain radiomic data for one technology (CT or MR or PET) in a selected disease condition i.e. strictly GI cancers (Esophageal/Gastric  cancer, Colorectal cancer, Pancreatic cancer); 

- to analyze radiomic RESULTS (for diagnosis, staging, prognosis, treatment monitoring, etc.) of all technologies for one selected disease condition; 

- to analyze literature data on radiomics (methods and results) with one technology (CT or MR or PET) in one cancer 

- to analyze  literature data on radiomics to assess BIOLOGY (i.e. Tumor differentiation and proliferation, tumor vascular invasion, etc) of one cancer type

Author Response

  1. First of all, GI cancers are a very heterogeneous group of tumors and especially liver cancer and NETs can't be mixed with gastric, pancreatic, or stromal and anal cancer.

Author response: Thank you for this comment. The article is extensively re-structured as per the Editor’s suggestion. We believe that the sections currently help differentiate these GI cancers from one another clearly. Also, there have been many valuable machine learning/radiomics based publications on liver and NET cancers and we believe the reader would benefit tremendously from these sections.

  1. All the imaging techniques are put together in the article without any consideration for the specific characteristics and limits in radiomic analysis for each one. For example, the problems of PET and CT or MR in extracting radiomic features are quite different. Especially PET is approached only superficially without consider inherent problems and specific aspects due to the type of radiopharmaceuticals used for some cancers (i.e. DOTATOC for NETs) and the use of percentage threshold of the maximum SUV to delineate tumor.

Author response: Thank you for the suggestion. We have added these limitations and cited relevant papers into the manuscript for increased thoroughness.

"Challenges with PET arise in the standardization of PET protocols across and even with-in institutions. Inherent issues with PET protocols arise given the nature of imaging ac-quisition, as a multitude of factures can include the standard uptake value. These may be physiologic, including patient motion, inflammation, or blood glucose levels. They may also be technical, including differences in calibration threshold, synchronization, injection time, and method of delivery. Specific to GI cancers, data acquisition may be limited by radiopharmaceuticals used for certain GI cancers (e.g. DOTATOC for NETs), in addition to the percentage threshold of the maximum standard uptake value used to delineate the tumor of interest. Figure 1 demonstrates the flowchart of application of AI in radiology for GI cancers.”

  1. Moreover, the technical aspects of radiomic features extraction are considered all together as the same thing i.e. algorythms, machine Learning and AI, but they are not.  

Author response:  We have reflected on this comment. In section 2 the techniques are distinctively discussed such as section 2.3 where the paragraphs are divided into manual engineered vs. deep learning etc.

  1. Finally, future applications are not investigated in the paper as you write in the abstract and in the introduction.

Author response: Thank you for this comment. Please refer to “Section 5: Future Challenges and Opportunities.” We have expanded on this per your comment.

So that, if you would revise the article it's necessary to radically change strategy. Some examples are given below: 

  • to analyze only METHODS (segmentation, Data acquisition and curation, features extraction) to obtain radiomic data for one technology (CT or MR or PET) in a selected disease condition i.e. strictly GI cancers (Esophageal/Gastric  cancer, Colorectal cancer, Pancreatic cancer); 
  • to analyze radiomic RESULTS (for diagnosis, staging, prognosis, treatment monitoring, etc.) of all technologies for one selected disease condition; 
  • to analyze literature data on radiomics (methods and results) with one technology (CT or MR or PET) in one cancer 
  • to analyze  literature data on radiomics to assess BIOLOGY (i.e. Tumor differentiation and proliferation, tumor vascular invasion, etc) of one cancer type

Author response: Thank you for this comment and suggestions. We have significantly re-structured and accordingly incorporated the required changes. Also, we have added the methods and biology were available. The content of each section is categorized based on modalities. All CT based publications are grouped together followed by MR based studies.

We believe the current layout achieves this in a way the reviewer believes important as well by discussing diagnosis/staging followed by prognosis/treatment.

Reviewer 2 Report

This manuscript gives an overview over recent studies evaluating the potential of radiomics in gastrointestinal cancer. It therefore remains rather superficial, while giving a broad overview over the topic. There are, however, some major and minor points that should be addressed before publication. Furthermore, some language editing (also regarding tenses) is needed. 

Abstract line 17 – change classify to „classification“

Line 32 – Please include an example of a screening test

Line 71: please add some examples that influence quantitative values

Line 161: doesn’t fit with the introduction (cause of death, gastric cancer)– please clarify in the text

Not only radiomics included – focus on radiomics or change title to “machine-learning”

Liver Cancer: rewrite the first paragraph – the introductory sentence focuses on other malignancies, the presented study only on differentiating metastases, hemangioma and HCC

Consider including the paragraph on neuroendocrine tumors in the pancreatic cancer section and you should move the paragraph on “anal cancer” to colorectal cancer, especially as it covers rectal cancer

Line 268: please add a reference as you talk about “others”

line 330: “Metastatic malignancies” – refers to CRC, so change to e.g. “CRC metastases”

Repeatedly you mention CT or MRI as possibility to diagnose/classify/stage cancer, but refer to a ML approach of imaging interpretation – please clarify, as of course cross-sectional imaging has a very central role in the management of cancer patients

Consider only mentioning that ML is also possible with US and delete the very few studies you found, and rather focus on PET, CT an MRI

Author Response

Dear Reviewer,

Thank you for the productive comments/suggestions. Please find our responses below.

  1. Abstract line 17 – change classify to „classification“

Author response: Thank you for this comment. We have revised this.

“Radiomics, a sub-field of computer vision analysis, is a bourgeoning area of interest especially in this era of precision medicine. In the field of oncology, radiomics has been described as a tool to aid in diagnosis, classification and categorize malignancies, and predict outcomes using various endpoints.”

  1. Line 32 – Please include an example of a screening test

Author response: Thank you for this comment. We have revised this

 “For some malignancies, screening programs have largely aided in early diagnosis of those at risk such as routine colon cancer screening.”

  1. Line 71: please add some examples that influence quantitative values

Author response:  Thank you for this comment. We have revised this in the manuscript.

  1. Line 161: doesn’t fit with the introduction (cause of death, gastric cancer)– please clarify in the text

Author response: Thank you. We have removed this line from section 3.2.

  1. Not only radiomics included – focus on radiomics or change title to “machine-learning”

Author response: Thank you for the suggestion. We agree with the reviewer and the title is changed to machine learning.

  1. Liver Cancer: rewrite the first paragraph – the introductory sentence focuses on other malignancies, the presented study only on differentiating metastases, hemangioma and HCC

Author response: Thank you for this comment. We have revised the introductory paragraph.

“Radiomics enables non-invasive differentiation of focal liver lesions, the most common primary hepatic malignancy being HCC but also including hemangioma and metastases. Radiomics signatures based on T2W-derived texture features of focal hepatic lesions can help classify hepatic hemangioma, hepatic metastases and HCC with good diagnostic performances (AUC: 0.83‐0.91).60 A previous study reported that primary liver tumors could be differentiated from metastatic lesions with an accuracy of 83% using 3D CNN features extracted from DWI images.61 Lastly, ultrasound image analysis can also classify benign and malignant focal liver lesions (AUC: 0.94) and malignant subtyping (AUC: 0.97).”

  1. Consider including the paragraph on neuroendocrine tumors in the pancreatic cancer section and you should move the paragraph on “anal cancer” to colorectal cancer, especially as it covers rectal cancer

Author response: Thank you for this suggestion. We have made extensive changes accordingly. The new subsections are: Gastric cancer, Colorectal cancer (including anal cancers), Pancreatic cancer (including neuroendocrine tumors), Liver cancers, GI stromal tumors.

  1. Line 268: please add a reference as you talk about “others”

Author response:  Thank you. We have added another reference.

  1. line 330: “Metastatic malignancies” – refers to CRC, so change to e.g. “CRC metastases”

Author response: Thank you for this suggestion. We have revised the title of this section to “CRC Metastases.”

  1. Repeatedly you mention CT or MRI as possibility to diagnose/classify/stage cancer, but refer to a ML approach of imaging interpretation – please clarify, as of course cross-sectional imaging has a very central role in the management of cancer patients

Author response: We have clarified that machine learning should be applied to cross sectional imaging (CT or MRI) to aid in diagnosing or classifying cancer. At this point, machine learning can only serve as adjunct to augment existing clinical imaging and aid in patients’ diagnosis and management. We have updated the manuscript to reflect this matter.

  1. Consider only mentioning that ML is also possible with US and delete the very few studies you found, and rather focus on PET, CT an MRI

Author response: We agree with the reviewer and have deleted the studies on ultrasound.

Reviewer 3 Report

The manuscript aims to assess current landscape of radiomics, which is one the area in medicine that has most benefited from the use of machine learning. Manuscript is comprehensive in its approach. Authors effort to summarize the radiomics in precision oncology is very commendable.

Minor comments:

1.     Manuscript has used many machines learning/AI lingo without any explanation, which may be not friendly with broad audience. Although terms such as AUC are very common terms in ML/AI, these terms must be mentioned with some explanation and why the AUC value is important.

2.     Authors could potentially include another flowchart that briefly visualizes how CNN, DNN, RF, KNN, etc. are related or different.

3.     It is recommended that authors include a proper citation. For e.g., Line 224

The AUC for the DL models outperformed the clinical model.1  224

Quickly skimming the paper, it was found that there was no mention of such information in that paper.

Major comment:

 In the abstract section, authors mentioned:

Finally, we discussed the existing challenges and limitations of radiomics in abdominal cancers and investigate future opportunities.

But the manuscript fails to elaborate on the challenges in the radiomics in precision oncology. It is recommended that authors elaborate on the current challenges with proper citation and authors' perspectives to overcome the challenges they have identified in the radiomics.

 Some recommended pointers:

Any AI/ML tool is as good as the training data it was provided and finding an optimum threshold to balance sensitivity and specificity is the key to the predictive performance in radiomics tools.

Patient data is highly heterogeneous because of age, sex, race, and demographics. A robust radiomics tool must account for highly heterogenous data for its generalizability.

False positives and overdiagnosis are some other concerns.

Testing the robustness of radiomics tools with prospective and retrospectives study.

Author Response

Dear Reviewer,

Thank you for the productive comments/suggestions. Please find our responses below.

Minor comments:

  1. Manuscript has used many machines learning/AI lingo without any explanation, which may be not friendly with broad audience. Although terms such as AUC are very common terms in ML/AI, these terms must be mentioned with some explanation and why the AUC value is important.

Author response: Thank you. We have made appropriate edits to the text.

  1. Authors could potentially include another flowchart that briefly visualizes how CNN, DNN, RF, KNN, etc. are related or different.

Author response:  Thank you for this comment. We appreciate this suggestion; In the manuscript, these automation modalities are cited so that readers may pursue them elsewhere.

Explaining the basics of machine learning is beyond the scope of this paper, we have included a very useful reference (https://www.sciencedirect.com/science/article/pii/S2211568420302461) to guide the readers on the basics of techniques.

  1. It is recommended that authors include a proper citation. For e.g., Line 224. The AUC for the DL models outperformed the clinical model.1  224 Quickly skimming the paper, it was found that there was no mention of such information in that paper.

Author response: Done. Thanks for pointing these out.

Major comment:

  1. In the abstract section, authors mentioned: “Finally, we discussed the existing challenges and limitations of radiomics in abdominal cancers and investigate future opportunities.” But the manuscript fails to elaborate on the challenges in the radiomics in precision oncology. It is recommended that authors elaborate on the current challenges with proper citation and authors' perspectives to overcome the challenges they have identified in the radiomics.

Some recommended pointers:

  • Any AI/ML tool is as good as the training data it was provided and finding an optimum threshold to balance sensitivity and specificity is the key to the predictive performance in radiomics tools.
  • Patient data is highly heterogeneous because of age, sex, race, and demographics. A robust radiomics tool must account for highly heterogenous data for its generalizability.
  • False positives and overdiagnosis are some other concerns.
  • Testing the robustness of radiomics tools with prospective and retrospectives study.

Author response: Thank you for these great suggestions. We have expanded Section 5: Future Challenges and Opportunities.

Redline: “Despite considerable progress, there remain some challenges in the application of radiomics to GI malignancies. First, any AI or ML tool is only as strong as the training data provided. Predictive performance for automated tools remains limited by lack of optimal thresholds necessary to balance sensitivity and specificity during data acquisition and curation. Similarly, given that patient data is highly heterogenous due to age, sex, race, and demographics, future algorithm and machine learning technologies must account for such variations. Furthermore, automated detection can still result in high rates of false positives. Conversely, radiomics holds potential to aid in potential overdiagnosis when used in conjunction with other clinical tools. It has been challenging to reach Dice similarity coefficients > mid 80% for segmentation accuracy using machine learning.  Lastly, even the best machine learning systems do not yet perform at the level of a radiologist, and it is apparent there is room for more advancement. Testing the robustness of radiomics tools with prospective and retrospective real life populations will be paramount to integrate such tools into clinical practice.”

Round 2

Reviewer 1 Report

it's can be fine

Reviewer 3 Report

Comments are addressed.